# LiDAR- and Radar-Based Robust Vehicle Localization with Confidence Estimation of Matching Results

**DOI:** 10.3390/s22093545

**Published:** 2022-05-06

**Authors:** Ryo Yanase, Daichi Hirano, Mohammad Aldibaja, Keisuke Yoneda, Naoki Suganuma

**Affiliations:** 1Advanced Mobility Research Institute, Kanazawa University, Kakuma-Machi, Kanazawa 920-1192, Ishikawa, Japan; amroaldibaja@staff.kanazawa-u.ac.jp (M.A.); suganuma@staff.kanazawa-u.ac.jp (N.S.); 2Graduate School of Natural Science and Technology, Kanazawa University, Kakuma-Machi, Kanazawa 920-1192, Ishikawa, Japan; d_hirano@stu.kanazawa-u.ac.jp; 3Institute for Frontier Science Initiative, Kanazawa University, Kakuma-Machi, Kanazawa 920-1192, Ishikawa, Japan; k.yoneda@staff.kanazawa-u.ac.jp

**Keywords:** localization, sensor fusion, autonomous driving

## Abstract

Localization is an important technology for autonomous driving. Map-matching using road surface pattern features gives accurate position estimation and has been used in autonomous driving tests on public roads. To provide highly safe autonomous driving, localization technology that is not affected by the environment is required. In particular, in snowy environments, the features of the road surface pattern may not be used for matching because the road surface is hidden. In such cases, it is necessary to construct a robust system by rejecting the matching results or making up for them with other sensors. On the other hand, millimeter-wave radar-based localization methods are not as accurate as LiDAR-based methods due to their ranging accuracy, but it has successfully achieved autonomous driving in snowy environments. Therefore, this paper proposes a localization method that combines LiDAR and millimeter-wave radar. We constructed a system that emphasizes LiDAR-based matching results during normal conditions when the road surface pattern is visible and emphasizes radar matching results when the road surface is not visible due to snow cover or other factors. This method achieves an accuracy that allows autonomous driving to continue regardless of normal or snowy conditions and more robust position estimation.

## 1. Introduction

In autonomous driving, localization is important for path planning, decision-making, operation, etc. Localization can be divided into two main types of methods. The first is satellite positioning using the Global Navigation Satellite System (GNSS), which is not available in places where radio waves do not reach, such as tunnels and mountain areas, and in high buildings, where multipath can decrease the estimation accuracy [1]. The second is map matching which estimates where the vehicle is located on a map by matching sensor data with a map related in advance. Commonly used features in autonomous driving include road surface patterns such as white lines and three-dimensional structures such as poles and buildings. In many cases, autonomous driving is achieved by combining satellite positioning and map matching, where satellite positioning is used to initialize and roughly determine location, and map matching is used for more precise position estimation. Thus, the challenge for safe autonomous driving is to increase the accuracy and robustness of map matching.

The 2007 Urban Challenge was the first demonstration of autonomous driving in an urban environment, and various universities and companies participated [2,3,4]. In this project, Levinson et al. developed a road surface pattern-based map matching technique using LiDAR (Light Detection Furthermore, Ranging) to achieve position estimation with RMS accuracy in the 10-cm range [5]. Since LiDAR has a ranging accuracy of a few centimeters, it can provide a detailed representation of road surface patterns and object shapes. For that reason, LiDAR-based map matching methods are mainly used in autonomous driving [6,7,8].

Many of the LiDARs used in research and development are high cost, thus one of the most important issues is to reduce the cost of LiDARs. As a low-cost sensor, MEMS (Micro Electro Mechanical Systems) mirror type LiDAR has been developed and is expected to be widely used in general vehicles. This sensor has a different field of view and resolution from the typical ones because of the different laser scanning mechanisms [9]. Kato et al. proposed and developed a localization method using MEMS mirror type LiDAR, considering its characteristics, and achieved a position estimation accuracy of 0.15 m [10].

Matching methods using 3-D point clouds instead of road surface patterns have also been proposed for LiDAR-based methods. Kato et al. have proposed a method for fast localization by using Normal Distribution Transform (NDT) scan matching to align 3-D point clouds [11]. Schaefer et al. have developed a method for extracting pole-like objects from a 3-D point cloud and aligning them with a map using these points [12].

However, LiDAR-based map-matching has issues to be solved to provide accurate position estimation for autonomous driving in various situations. For example, under heavy rain conditions, it may not be possible to correctly obtain the pattern and shape of the road surface due to laser reflection and scattering. Furthermore, in snowy environments, snow can change road shape and cover the road surface. Under such situations, the map and sensor data features can be significantly different, causing incorrect matching results, which decreases the estimation accuracy. The strategy to avoid this issue is to simply reject the matching results. Rejecting the results will suppress the influence of the environment, but if the problematic situation occurs over a certain long interval, the matching results will be continuously rejected during that section, which may increase the error. To solve this problem, other methods and sensors should be used in combination to ensure system redundancy.

Millimeter-wave radar is a candidate sensor for use in combination with LiDAR. A method based on the observation of objects such as poles and guardrails using millimeter-wave radar and matching them by their features has been proposed to estimate the position, achieving autonomous driving in urban areas during snowy weather [13]. Millimeter-wave radar has a lower estimation accuracy than LiDAR-based localization in terms of ranging accuracy, and quantitative evaluations have shown that the RSM accuracy is about 0.25 m.

Other methods include camera-based localization. As a monocular camera-based method, a technique for detecting road surface patterns such as lane marking and matching them with a map has been proposed [14]. A stereo camera-based method has been proposed that reconstructs a 3-D point cloud from disparities and applies scan matching for position estimation [15]. Brubaker et al. have proposed a visual odometry-based self-localization method using OpenStreetMap [16]. The unique feature of this method is that it can estimate locations without GNSS data. The trajectory of a vehicle can be obtained using visual odometry calculated from camera images, and the vehicle’s position can be determined by matching the trajectory with the road structure of a map. However, the accuracy of this method is in the range of a few meters, which is insufficient for safe autonomous driving standalone. In addition, cameras are not suitable for use with LiDAR because they cannot provide precise information in bad weather conditions, such as snow if their view is occluded.

Several methods have been proposed that use multiple different sensors in combination. Tsuchiya et al. have proposed a method that combines a camera and LiDAR using a map of road surface pattern and a 3-D point cloud map [17]. In the research, the difference between the angle of the detected white line and that on the map at the predicted position is used as the confidence level of the white line matching result. If the angular difference is greater than a threshold, it can be rejected. The confidence level of the NDT scan matching of the 3-D point cloud is determined based on the histogram of the Mahalanobis distance. If the percentage of point clouds with small Mahalanobis distances is less than a threshold, the matching result can be rejected. This method achieves an accuracy of less than 0.1 m in the vehicle lateral direction and about 0.3 m in the vehicle forward/backward direction. However, the method of setting thresholds for the difference between observations and predictions and the similarity of matching results can only reject obvious mismatches.

Lee et al. have proposed a self-localization method that combines a camera and LiDAR and sets a confidence level for each sensor’s observation and each map used for map matching [18]. For LiDAR, NDT matching of 3D point clouds is performed, and for the camera, white line matching is performed. The confidence level of the LiDAR is set based on the percentage of the number of objects determined to be stationary objects out of the total number of objects detected by clustering among the observed point clouds. When the number of stationary objects is high and the number of moving objects is low, the confidence level of the observation is increased. The confidence level of the LiDAR map is increased when the map contains many objects of a certain height and decreases when there are few. Camera observation confidence is calculated based on the number of white lines detected, and camera map confidence is calculated in the same way as for observations.

Besides, a system using learning by Convolution Neural Network (CNN) has been developed as a method for estimating the confidence level of matching [19]. This method uses a particle filter to estimate position, and at the same time determines whether the result is in the correct position. If it determines that the estimation has failed, it resets the localization to recover the correct position. The purpose of our research in this paper is to reject incorrect matching results and prevent a loss of estimation accuracy. Therefore, the above method is a different approach from the requirements of this study.

In this paper, we propose a localization method that combines LiDAR and millimeter-wave radar. During normal conditions when the road surface pattern is visible, the matching results of LiDAR are emphasized for highly accurate position estimation. On the other hand, when the road surface is not visible due to heavy rain or snowfall, radar matching results are emphasized to provide position estimation that enables continued automated driving, even if the accuracy is lower than that based on LiDAR. This provides weather-independent localization. For this, it is necessary to determine whether the road surface pattern is visible or not, and solving this problem is critical to realizing the system. For this purpose, we propose a confidence estimation method based on deep learning. The calculated confidence level is used as a weight when integrating LiDAR matching results, thereby changing the influence of LiDAR results. We also implement the proposed method and evaluate its effectiveness in autonomous driving by verifying its accuracy using actual driving data in a real environment.

## 2. Proposed Method

### 2.1. Vehicle and Sensor Configuration

As a prerequisite, the vehicle and sensor configuration used in this study are described. The test vehicle and sensor configuration are shown in Figure 1. The vehicle is equipped with an Applanix POS-LV220 GNSS/INS(Global Navigation Satellite System/Inertial Navigation System) that acquires position (latitude, longitude, and altitude) and attitude angle (roll, pitch, and yaw) at a frequency of 100 Hz. In this study, positions, where off-line post-processing corrections were performed on the POS-LV220 data, are used as ground truth in the evaluation.

A camera is mounted on the front-side glass. This camera is used for traffic signal recognition in autonomous driving. It is not used for self-position estimation, but it provides important information about the conditions of the location when analyzing data, such as when an issue is found.

The LIDAR sensor is a Velodyne VLS-128 Alpha Prime, which has 128 laser transmitting and receiving sensors. This sensor is capable of measuring three-dimensional distance omnidirectional horizontally and can acquire data at a frequency of 10 Hz.

A total of nine 77 GHz millimeter-wave radars are mounted on the front and rear bumpers, and the mounting positions and directions of the sensors are shown in Figure 2. Each radar acquires the distance, angle, and relative velocity of an object at a frequency of 20 Hz.

### 2.2. Overview

The proposed method consists of two steps: dead-reckoning and map matching. First, a rough position is estimated by dead reckoning using the velocity vector obtained from GNSS/INS. A dead-reckoning position can be calculated by integration of velocity vectors over time. Let a velocity vector in time t − 1 be $v_{t-1}$, a time period be Δt and the dead-reckoning position in time t − 1 be $x_{t-1}^{\mathrm{DR}}$, the dead-reckoning position $x_{t}^{\mathrm{DR}}$ in time *t* is calculated by Equation (Equation 1).
(1)$$\begin{array}{cc} x_{t}^{\mathrm{DR}} & =x_{t-1}^{\mathrm{DR}}+v_{t-1}\Delta t \end{array}$$

The dead-reckoning error (hereinafter, referred as to “offset”) increases proportionally to the driven distance from the initial position because of the error accumulation. Therefore, the proposed method estimates the offset $\Delta x_{t}^{\mathrm{DR}}$ in order to compute the actual position $x_{t}^{v}$ of the vehicle as in (Equation 2).
(2)$$\begin{array}{cc} x_{t}^{v} & =x_{t}^{\mathrm{DR}}+\Delta x_{t}^{\mathrm{DR}} \end{array}$$

Figure 3 shows the flowchart of the proposed localization framework. When data is acquired from LiDAR and radar, the data is matched with a map for each sensor. Matching is performed on an image basis, resulting in a correlation distribution. The probability distribution of the offset is updated by a histogram filter using the correlation distribution as the likelihood of the offset.

### 2.3. Map-Matching

This section explains map-matching for LiDAR and radar, respectively. The map matching procedure is performed as follows:1.Project the sensor data onto a two-dimensional plane and create an image (observation image).2.Cut out the map image corresponding to the vehicle position.3.Calculate the correlation distribution by template matching.

The only difference for each sensor is the part of generating the observation image (procedure 1), while procedures 2 and 3 are common to both sensors.

#### 2.3.1. LiDAR Observation Image

The LiDAR observation image is obtained by mapping the laser reflectivity of the road surface onto a two-dimensional image. The center pixel of the observation image is always the vehicle position estimated by dead-reckoning because position estimation can be performed without a large increase in error if dead-reckoning is limited to short intervals of a few seconds. Let the vehicle position in absolute coordinate system be $x_{t}^{v}\;=\;{\left[\begin{array}{cc} x_{t}^{v} & y_{t}^{v} \end{array}\right]}^{T}$, a rotation matrix representing the vehicle attitude (roll, pitch and yaw) be R, and the position of the LiDAR observation point in the vehicle coordinate system be $x_{\mathrm{obs}}^{\mathrm{VCS}}$ the position $x_{\mathrm{obs}}^{\mathrm{ACS}}$ of an observation in the absolute coordinate system is calculated by Equation (Equation 3).
(3)$$\begin{array}{c} x_{\mathrm{obs}}^{\mathrm{ACS}}\;=\;R\;x_{\mathrm{obs}}^{\mathrm{VCS}}\;+\;x_{t}^{v}. \end{array}$$

To consider quantization errors in mapping, the position of the center pixel in the absolute coordinate system is computed from Equation (Equation 4). Then, a point cloud of the road surface is projected onto the image. An arbitrary LiDAR observation point is mapped to a pixel according to the relationship shown in Figure 4, which is obtained by Equation (5).
(4)$$\begin{array}{cc} \left[\begin{array}{c} x_{\mathrm{center}}\\ y_{\mathrm{center}} \end{array}\right] & =\left[\begin{array}{c} D_{\mathrm{res}}\;\cdot \;\mathrm{round}\left(x_{t}^{v}/D_{\mathrm{res}}\right)\\ D_{\mathrm{res}}\;\cdot \;\mathrm{round}\left(y_{t}^{v}/D_{\mathrm{res}}\right) \end{array}\right] \end{array}$$
(5)$$\begin{array}{cc} \left[\begin{array}{c} u_{\mathrm{obs}}\\ v_{\mathrm{obs}} \end{array}\right] & =\left[\begin{array}{c} \mathrm{round}\left(\left(x_{\mathrm{obs}}^{\mathrm{ACS}}\;-\;x_{\mathrm{center}}\right)/D_{\mathrm{res}}\right)\;+\;u_{\mathrm{center}}\\ \mathrm{round}\left(\left(y_{\mathrm{center}}\;-\;y_{\mathrm{obs}}^{\mathrm{ACS}}\right)/D_{\mathrm{res}}\right)\;+\;v_{\mathrm{center}} \end{array}\right], \end{array}$$
where $D_{\mathrm{res}}$ is the resolution of the image, and $D_{\mathrm{res}}\;=\;0.125$ [m/pixel]. Since the thickness of a typical white line is 0.15 [m], the resolution is set to 0.125 [m].

If multiple points are projected to the same pixel, the average reflectivity is assigned to that pixel. Since a single frame by itself provides a sparse observation image, a dense observation image is generated by integrating frames over several seconds. For multi-frame integration, the position of the vehicle at the time of each frame is used to generate the observed image, and the overlap among images is determined by calculating the amount of motion between frames based on the vehicle velocity. For pixels that overlap between frames, the average reflectivity is assigned in the same way as in the single-frame case.

#### 2.3.2. Radar Observation Image

Since millimeter-wave radar has lower ranging accuracy and angular accuracy than LiDAR, the mapping needs to consider the sensor characteristics. The probability of the existence of a stationary obstacle, including the observation error, is projected onto the observation image as shown Figure 5. A projected pixel of the observed object can be calculated in the same way as for LiDAR. Next, consider the error ellipse of the observation position that extends around this pixel. This error ellipse can be computed from elements in the covariance matrix of observation information, especially concerning position only. The error model is based on the study in [20].

The size of the error ellipse can be defined from the chi-square distribution by determining the number of dimensions and the significance level. By setting the number of dimensions to two (direction and orientation) and the significance level to 0.05, the size of the error ellipse is calculated to be 5.99, and the region where the observation information is considered to be obtained with 95[%] probability can be determined. The computed probabilities are converted to pixel values to generate an observation image. Furthermore, when the frames are accumulated over several seconds, a binary Bayes filter [21] is used to update the existence probabilities of the objects.

#### 2.3.3. Pre-Defined Map

The map used for matching is created offline by driving the target area once and collecting sensor data. Based on the GNSS/INS position, LiDAR and radar data can be mapped onto an image using the method described above to create a map for each sensor. The region around the estimated position $x_{t}^{v}$ of the vehicle is cut out and used in the matching process. Examples of LiDAR and radar map images are shown in Figure 6 and Figure 7.

#### 2.3.4. Template Matching

For each sensor, a correlation distribution is computed by template matching between the observed image and the map image. As shown in Figure 8, the correlation distribution is the distribution of similarity when two images are overlapped by a shift of (ΔU,ΔV). This method uses ZNCC (Zero-means Normalized Cross-Correlation) as the similarity. The correlation value is highest at the position corresponding to the vehicle’s location on the map image. It can be regarded as a likelihood distribution of offsets. Since the observation image is created centered on the vehicle origin and the map image is cropped around the estimated position of the vehicle, the offset (Δx,Δy) is a shift in the center between the two images. Let $\left(C_{u}^{\mathrm{map}},C_{v}^{\mathrm{map}}\right)$ be the center pixel of the map image and $\left(C_{u}^{\mathrm{obs}},C_{v}^{\mathrm{obs}}\right)$ be the center pixel of the observation image, the relationship between the offset (Δx,Δy) and the shift (ΔU,ΔV) can be expressed by the following Equation (Equation 6).
(6)$$\begin{array}{cc} \left[\begin{array}{c} \Delta x\\ \Delta y \end{array}\right] & =D_{\mathrm{res}}\left[\begin{array}{c} C_{u}^{\mathrm{obs}}\;+\;\Delta U\;-\;C_{u}^{\mathrm{map}}\\ C_{v}^{\mathrm{map}}\;-\;\left(C_{v}^{\mathrm{obs}}\;+\;\Delta V\right) \end{array}\right] \end{array}$$

However, this assumes that the observation image has the same appearance as the map image, and if the road surface is covered by snow, the pattern will be different from the map image and the correlation value may not be high at the correct position. Therefore, it is important to estimate the confidence level that the matching result is correct for LiDAR template matching.

### 2.4. Confidence Estimation of LiDAR Matching Result

This section explains the confidence estimation of LiDAR matching results. As mentioned above, if the road surface pattern of the observation image is different from that of the map image at the same location, high correlation values may not be calculated at the correct location. In fact, during snowfall or heavy rain, the correlation peaks often appear in incorrect locations due to the lack of visibility of the road surface pattern. In other words, if the observation image does not have the same road surface pattern as the map, the reliability of the matching result can be regarded as low. This binary classification problem takes two images as input and determines whether the observation image has the same road surface pattern as the map image.

A classifier is created using a convolutional neural network (CNN), which is often used for image classification. Figure 9 shows the structure of the classifier for confidence estimation. Here, the architecture is modeled in reference to the method used to calculate the similarity between two images [22]. For each input image, convolution layers and max-pooling layers are used to extract features. The computed features are concatenated into a single vector. The feature is learned using fully-connected layers, ReLU activation functions, and dropout layers, and finally, a sigmoid function converts the computed result to a value between 0 and 1 and outputs it as a confidence level.

### 2.5. Probability Update and Offset Estimation

The correlation distributions computed by the map matching are integrated into a single probability distribution over time to integrate the two types of observation results, LiDAR, and radar. Since LiDAR provides more accurate observations than radar, a probability distribution is generated that is significantly influenced by LiDAR observations under normal conditions. The influence of LiDAR can be changed depending on the situation by giving the confidence level of the matching result as a weight when updating the probability distribution using LiDAR, and it is also possible to generate a probability distribution that relies on the radar.

When the correlation distribution is obtained by map matching, the likelihood distribution of the offset at present is created as explained above. It would be better if the likelihood distribution had a peak value and only one expected position for the offset, but if there are few features in the observation image, the shape of the distribution becomes ambiguous and the peak may not always be at the correct position. Therefore, this method processes the likelihood distribution over time and integrates the LiDAR and radar matching results into a single posterior probability distribution. This process provides a stable and reliable distribution for estimating offsets.

The posterior probability distribution represents where the offset is most likely to be located. The computation of the posterior probability distribution consists of two processes: time update and observation update. Time update predicts the distribution at the current time based on the posterior probability distribution of the previous frame and creates a prior probability distribution based on the amount of movement of the vehicle. The observation update combines the prior probability distribution with the likelihood distribution at the current time to generate a posterior probability distribution. The posterior probability distribution is updated by a binary Bayes filter based on Bayes’ rule.

#### 2.5.1. Time Update

This section describes the time update that predicts the prior probability distribution $P_{t/t-1}$ at the current time *t* based on the posterior probability distribution $P_{t-1/t-1}$ generated at the previous time t − 1. The left subscript *t* of the distribution $P_{t/t-1}$ denotes the distribution at time *t*, and the right subscript t − 1 denotes the distribution calculated using the likelihood distribution up to time t − 1. When predicting the prior probability distribution at the current time from the prior probability distribution at the previous time, considering that the vehicle is moving, its position is calculated from the GNSS/INS measurements, which have errors, and is modeled as a Gaussian distribution based on that position. Assuming that the GNSS/INS errors are similar in the longitudinal and lateral directions of the vehicle, the probability in the prior probability distribution can be expressed by (Equation 7).
(7)$$\begin{array}{cc} P_{t/t-1} & =\sum_{i,j}P_{t-1/t-1}\left(i,j\right)\exp\left(-\frac{{\left(i-\Delta x\right)}^{2}+{\left(j-\Delta y\right)}^{2}}{2\sigma_{t}^{2}}\right), \end{array}$$
where $\sigma_{t}=\alpha v_{t-1}\left|\Delta t\right|$ represents the error variance, Δt is the elapsed time between time *t* and t−1, and α is a constant. The error covariance is determined using the vehicle movement per unit time with vehicle velocity $v_{t-1}$.

#### 2.5.2. Observation Update for LiDAR

This section explains how to transform the results of LiDAR frame matching into a probability distribution and how to update the posterior probability distribution. Gamma correction and normalization of the correlation distribution are performed as preprocessing. This process is based on previous studies [5,7] and is applied to increase the difference in likelihood. The gamma correction is empirically computed to the fourth power of the correlation value, and negative values are not processed and assigned the value of 0. For the exponential part of the gamma correction, the value is empirically determined from the correlation values obtained by template matching. The correlation distribution ${\mathrm{CORR}}_{\mathrm{LiDAR}}$ of LiDAR generated by this preprocessing is shown in (Equation 8).
(8)$$\begin{array}{cc} {\mathrm{CORR}}_{\mathrm{LiDAR}} & =\max{\left(\mathrm{ZNCC},0\right)}^{4}/{\mathrm{CORR}}_{\max}, \end{array}$$
where ${\mathrm{CORR}}_{\max}$ is the maximum value in the raw correlation distribution.

Next, the correlation values are transformed into probabilities, and the distribution obtained here is regarded as the likelihood distribution at the current time. The binary Bayes filter increases the posterior probability if the observation probability (likelihood) is greater than 0.5 and decreases the posterior probability if it is less than 0.5. Therefore, as shown in Figure 10, we transform the correlation so that the probability is 0.5 if the correlation value is at a given threshold ${\mathrm{CORR}}_{\mathrm{th}}$. In addition, the range of probability is adjusted according to the confidence level $g_{c}$ of the LiDAR matching results estimated by CNN. The likelihood ${\mathrm{CORR}}_{t,\mathrm{LiDAR}}\left(\Delta x\right)$ of offset Δx is calculated by (Equation 9).
(9)$$\begin{array}{cc} {\mathrm{CORR}}_{t,\mathrm{LiDAR}}\left(\Delta x\right) & =\left\{\begin{array}{cc} \frac{g_{c}}{2}\frac{{\mathrm{CORR}}_{\mathrm{LiDAR}}\left(\Delta x\right)\;-\;{\mathrm{CORR}}_{\mathrm{th}}}{1\;-\;R_{\mathrm{th}}}\;+\;0.5 & \left({\mathrm{CORR}}_{\mathrm{LiDAR}}\left(\Delta x\right)\;\geq \;R_{\mathrm{th}}\right)\\ 0.5-\frac{g_{c}}{2}\frac{{\mathrm{CORR}}_{\mathrm{LiDAR}}\left(\Delta x\right)}{{\mathrm{CORR}}_{\mathrm{th}}} & \left({\mathrm{CORR}}_{\mathrm{LiDAR}}\left(\Delta x\right)\;<\;{\mathrm{CORR}}_{\mathrm{th}}\right) \end{array}\right. \end{array}$$

Finally, the posterior probability distribution is computed from the prior probability distribution and the likelihood distribution. Binary Bayesian filters can be computed by the simple addition of odds values, using a log-odds representation of probability. The log-odds *L* are expressed as L = ln(p/1 − p) for probability *p*. Let the log-odds of the prior, likelihood distribution, and posterior probabilities be $L_{t/t-1}\left(\Delta x\right)$, $L_{t,\mathrm{LiDAR}}\left(\Delta x\right)$, and $L_{t/t}\left(\Delta x\right)$, respectively, then the probability can be updated using the following Equation (Equation 10).
(10)$$\begin{array}{cc} L_{t/t}\left(\Delta x\right) & =L_{t/t-1}\left(\Delta x\right)+k_{\mathrm{LiDAR}}L_{t,\mathrm{LiDAR}}\left(\Delta x\right), \end{array}$$
where $k_{\mathrm{LiDAR}}$ is a gain and is set to a constant value. In this paper, $k_{\mathrm{LiDAR}}=1$ is set experimentally.

#### 2.5.3. Observation Update for Radar

For radar, the probability distribution is updated as in the case of LiDAR. Because of the lower ranging accuracy of radar compared to LiDAR, the likelihood distribution is calculated to consider the instability of the measurement. Gamma correction is performed by cubing the correlation value as in (Equation 11). As in the case of LiDAR, the exponential part of the gamma correction is determined empirically from the correlations. The conversion to likelihood is calculated so that the range of probability becomes small, as shown in (12).
(11)$$\begin{array}{cc} {\mathrm{CORR}}_{\mathrm{Radar}} & =\max{\left(\mathrm{ZNCC},0\right)}^{3}/{\mathrm{CORR}}_{\max}, \end{array}$$
(12)$$\begin{array}{cc} {\mathrm{CORR}}_{t,\mathrm{Radar}}\left(\Delta x\right) & =0.4({\mathrm{CORR}}_{\mathrm{Radar}}-0.125)+0.5, \end{array}$$
where ${\mathrm{CORR}}_{\max}$ is the maximum value in the raw correlation distribution.

The posterior probability distribution is calculated by adding the log odds as well as LiDAR. Let the log-odds of the prior, likelihood distribution, and posterior probabilities be $L_{t/t-1}\left(\Delta x\right)$, $L_{t,\mathrm{Radar}}\left(\Delta x\right)$, and $L_{t/t}\left(\Delta x\right)$, respectively, then the probability can be updated using the following Equation (Equation 13).
(13)$$\begin{array}{cc} L_{t/t}\left(\Delta x\right) & =L_{t/t-1}\left(\Delta x\right)+k_{\mathrm{Radar}}L_{t,\mathrm{Radar}}\left(\Delta x\right), \end{array}$$
where $k_{\mathrm{Radar}}$ is a gain. In this paper, $k_{\mathrm{Radar}}=0.045$ is set experimentally.

### 2.6. Offset Calculation

In the posterior probability distribution, a pixel with a higher value represents a higher likelihood of being a true offset. First, pixels with a probability higher than a given threshold are extracted for peak detection. In this method, the threshold is set to 75[%]. Next, the offset is computed by weighted mean of the extracted pixels. Let the probability distribution after peak extraction be $P^{*}$, the offset $\Delta x_{t}={\left[\begin{array}{cc} \Delta x_{t} & \Delta y_{t} \end{array}\right]}^{T}$ at the present time is calculated by (Equation 14).
(14)$$\begin{array}{cc} \Delta x_{t} & =\frac{\sum_{\Delta x}\left(P^{*}\left(\Delta x\right)\Delta x\right)}{\sum_{\Delta x}(P^{*}\left(\Delta x\right)}\\ \Delta y_{t} & =\frac{\sum_{\Delta x}\left(P^{*}\left(\Delta x\right)\Delta y\right)}{\sum_{\Delta x}(P^{*}\left(\Delta x\right)} \end{array}$$

In practice, a smoother offset estimation can be performed by applying a low-pass filter.

## 3. Results

In this section, two main evaluations are conducted. First, the effectiveness of the proposed confidence estimation model for CNNs is verified. Next, the accuracy of localization using vehicle driving data is verified. The estimation accuracy of the proposed method is compared with that of the conventional LiDAR-based and radar-based methods.

### 3.1. Accuracy of Confidence Estimation

This section presents the results of the confidence estimation of the proposed model with training data. Train and validation datasets were created from data previously measured by the test vehicle. Observation and map images with visible road surface patterns were given the label “LaneLine”, while those with no visible patterns were given the label “NoLine”. The training was performed so that the model outputs 1 if the pattern is visible and 0 otherwise. Examples of training data are shown in Figure 11.

Table 1 shows the number of datasets created. As a component of the data given the label “NoLine”, approximately 20% of the images show no white lines because the road surface is wet (in heavy rain condition), 50% of images show the road surface completely covered with snow, and the remaining 30% of images show the road surface partially covered with remaining snow. When assigning labels, “LaneLine” is assigned to images that can be judged to be white lines by the human eye, and “NoLine” is assigned to those that are not. In this study, images that are difficult to judge are not included in this dataset. Data for snow-covered road surfaces were collected in January 2021, and data for dry road surfaces were collected in May 2021.

The Optimizer is Adam and the loss function is BCE loss [23]. The learning rate was determined to be 0.000001, the number of epochs to be 1000, and the batch size to be 32. The data were randomly flipped horizontally/vertically and randomly rotated in the range of −180 to 180 degrees for data augmentation. The GPU used for training was an NVIDIA GeForce GTX 1070, and training took about 40 h. Figure 12 shows the training results of the model obtained with the above dataset and training conditions. Accuracy and loss for both Train/Validation have converged, and the learning process is progressing. Table 2 shows the Loss and Accuracy for the training and validation data at the end of training. Recall = 0.986 and Precision = 0.913 were obtained from the confusion matrix shown in Table 3. The results show that the system almost always correctly determines whether the road surface pattern is visible or not.

### 3.2. Localization Accuracy

Next, we evaluate the accuracy of position when confidence estimation is integrated into LiDAR and radar-based localization. By applying post-processing corrections to GNSS/INS data, a very accurate position can be obtained. The post-processing position was used as the ground truth, and the errors of the estimated position in longitudinal and lateral directions of the vehicle were evaluated. For evaluation, experiments were conducted in two different environments: a dry road surface condition and a snow-covered road surface condition. The evaluation route is shown in Figure 13. The route is approximately 4.6 [km] long and covers the urban area of Kanazawa, Ishikawa, Japan.

#### 3.2.1. Dry Road Surface Condition

Table 4 shows the RMS and maximum errors, and Figure 14 and Figure 15 show the longitudinal and lateral errors for each method. Table 4 shows that the RMS errors for all methods are within a similar range, and the radar-based position estimation appears to be more accurate when compared in terms of maximum error.

However, the graphs in Figure 14 and Figure 15 provide another perspective from which to evaluate accuracy. The horizontal axis of these graphs shows the amount of error, and the vertical axis shows the ratio of data within the error to the total data. For example, focusing on Figure 15, the percentage of the lateral error within 0.2 [m] is about 84% for the standalone radar method, while it is about 98% for the standalone LiDAR and the proposed method. In other words, the standalone LiDAR method has more data for highly accurate position estimation than the radar method. Furthermore, the ratio of the proposed method is equivalent to that of the LiDAR results. Therefore, on dry road surfaces, both longitudinal and lateral accuracies of the proposed method are comparable to those of LiDAR-based localization.

#### 3.2.2. Snow-Covered Road Surface Condition

Table 5 show the RMS and maximum errors, and Figure 16 and Figure 17 show the longitudinal and lateral errors for each method on snow-covered road surface. Table 5 shows that the errors of the standalone LiDAR method are much larger than those of the radar one, which means that the position estimation is incorrect. Comparing the cumulative ratio of errors, the LiDAR method has a small percentage of data within a small error. The RMS and maximum errors of the proposed method are comparable to those of the radar method, and the charts of the ratio of errors overlap with the radar results in both vertical and horizontal directions. Therefore, it can be seen that in snowy environments where the road surface is not visible, the accuracy of position estimation is equivalent to that of a standalone radar system. In addition, the results of the proposed method show that more than 90% of the data are within 0.5 [m] of the error. Considering that a typical road width is 4 [m], a 0.5 [m] deviation corresponds to about 1/8 of the lane width. This error can cause, for example, a vehicle to drive close to the edge of the road when it is supposed to be traveling in the center of the lane. Although it cannot be guaranteed to be safe, the accuracy of the system is considered sufficient to achieve autonomous driving, assuming that the system can be covered by object recognition and path planning technologies.

These results provide evidence that position estimation can be performed as intended, focusing on LiDAR matching results when the road surface is visible, and focusing on radar results when it is not.

#### 3.2.3. Processing Time

To evaluate the computational complexity, we measured the processing time. As a result, the average computation time per frame was 76.0 ms (standard deviation: 7.0 ms) and the maximum computation time was 110.7 ms. Since 99.9% of the measured data was less than 100 ms and within 0.1 s, which is the data acquisition cycle of LiDAR, this method can be practically used in real-time.

## 4. Disscusison

The evaluation results in the previous section concluded that the system can be used for autonomous driving, thus we conducted a test on a public road. The experiment was conducted in Abashiri City, Hokkaido, Japan, an environment with a lot of snow. This area allows for a long-distance route and includes various scenes such as urban areas and country roads. The relatively low traffic volume also made it possible to experiment with sufficient safety considerations. The actual autonomous driving can be checked in the video at the following URL: https://www.dropbox.com/s/ox1cenosdgpadiy/test_abashiri.mp4 (accessed on 9 February 2022). As a result, autonomous driving was successful for the most part, but some issues were found. In this section, we discuss these issues.

### 4.1. Incorrect Estimation of Confidence Level

In sections where the road surface is partially covered, a high confidence level is calculated, causing a problem in which incorrect matching results are used for offset estimation, resulting in large positional deviations. Figure 18 shows an actual example of a false estimation. In the training of confidence estimation, the test data was provided with images that were relatively easy to determine, so that cases in which the road surface was partially hidden were omitted from the data. Therefore, adding such images to the training data and re-learning them is the first solution to this problem. However, there is a concern that even if the road surface is visible, the confidence level may be low if the road surface pattern is partially blurred. It is also very difficult to determine the criteria for the correct labeling, i.e., how much region of the road surface that is hidden should be labeled as “NoLane”.

Another solution is not a binary classification problem, but a method to determine pixels in the observation image where occlusion occurs or where features different from those in the map appear. This allows matching to be performed while ignoring pixels with features that differ from those in the map, thereby improving matching accuracy.

### 4.2. Mismatching of Radar Images

In snowy environments, position estimation relies on radar matching results, but there are situations where radar matching errors decrease accuracy. Originally, radar penetrates snow and can detect poles and other objects located behind it. However, if the amount of snow is heavy and compressed, the snow itself may be detected as an object. Figure 19 shows an example of radar mismatching. Edges that do not exist in the map image appear in the observation image. This causes the peak of the correlation distribution in the matching result to appear in the incorrect position, leading to incorrect offset estimation. As in the case of LiDAR, this problem is caused by the difference in features between the map image and the observation image, and such matching results should be rejected. In this study, we mainly considered confidence estimation of LiDAR matching results, but to develop a more robust system, it is necessary to introduce confidence estimation for radar results as well. Alternatively, other methods could be combined to improve system redundancy.

## 5. Conclusions

Map-matching methods based on LiDAR can provide highly accurate position estimation. However, in situations where the road surface is hidden, such as in snowy environments, the accuracy of position estimation is decreased because the features of the map and sensor data differ significantly, and matching results can be incorrect. To solve this problem, this paper proposed a localization system that uses both LiDAR and millimeter-wave radar, giving priority to LiDAR results when the road surface pattern is visible and radar results when the road surface pattern is not. By determining whether the road surface pattern is visible using deep learning, we estimate the confidence level of the LiDAR matching result, which in effect changes the influence of LiDAR on the position estimation. We conducted experiments under the condition that the road surface was visible and in snowy environments, and confirmed that the estimation accuracy was sufficient for autonomous driving under both conditions. To achieve a more robust system, it is necessary to increase the training data for confidence estimation and to estimate confidence levels for radar matching results. 

## Figures and Tables

**Figure 1 sensors-22-03545-f001:**
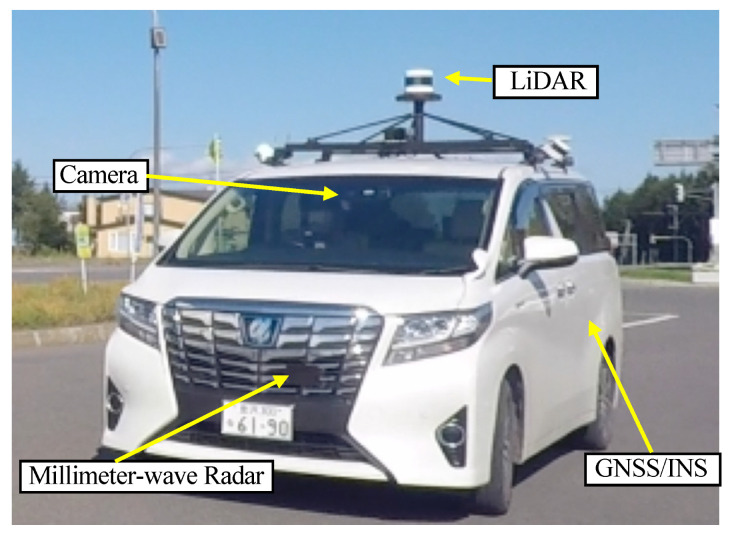
Test Vehicle.

**Figure 2 sensors-22-03545-f002:**
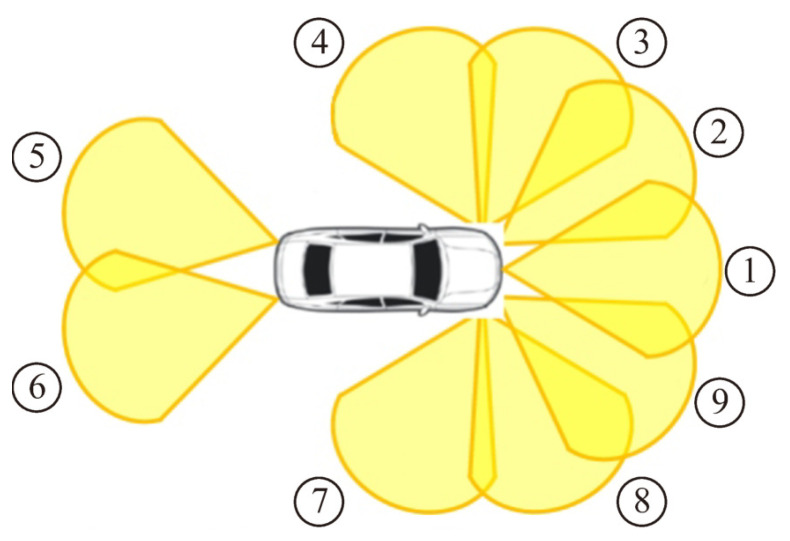
Sensor layout of millimeter-wave radars. The vehicle has nine radars. The numbers are sensor IDs.

**Figure 3 sensors-22-03545-f003:**
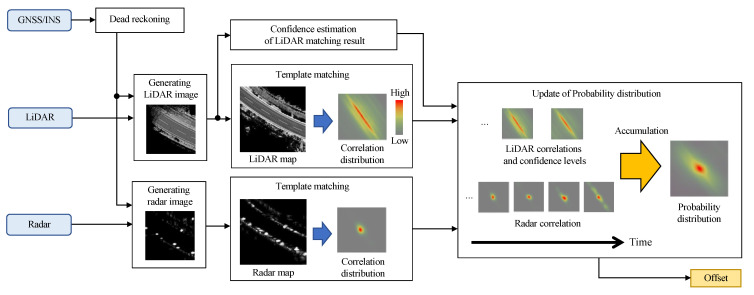
The overview of the localization framework.

**Figure 4 sensors-22-03545-f004:**
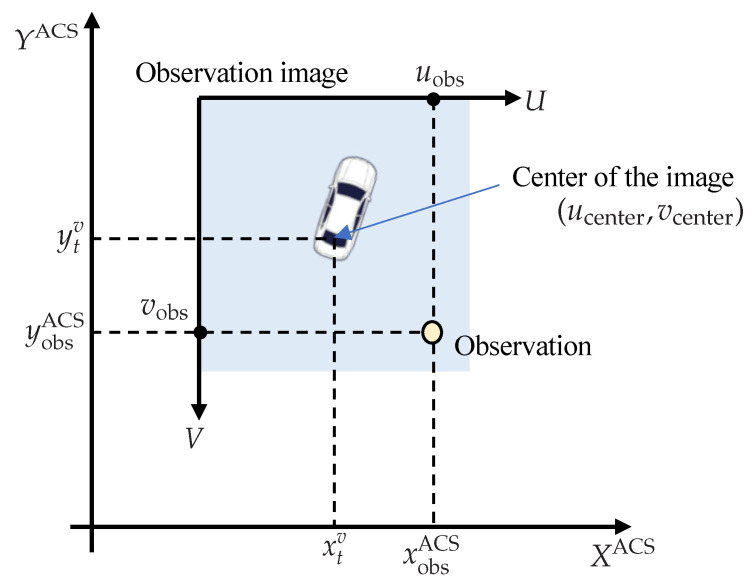
Mapping of a LiDAR observation to absolute coordinate system.

**Figure 5 sensors-22-03545-f005:**
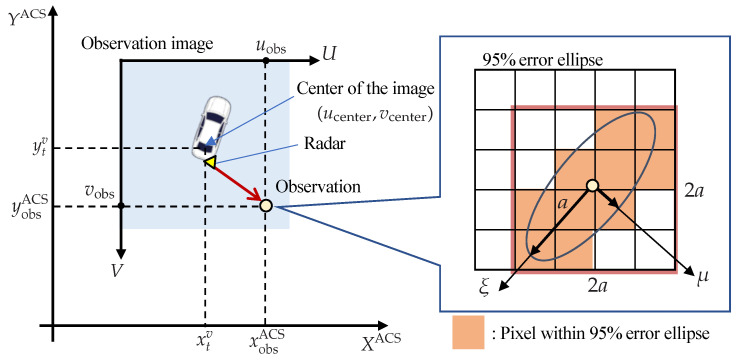
Mapping of a radar observation.

**Figure 6 sensors-22-03545-f006:**
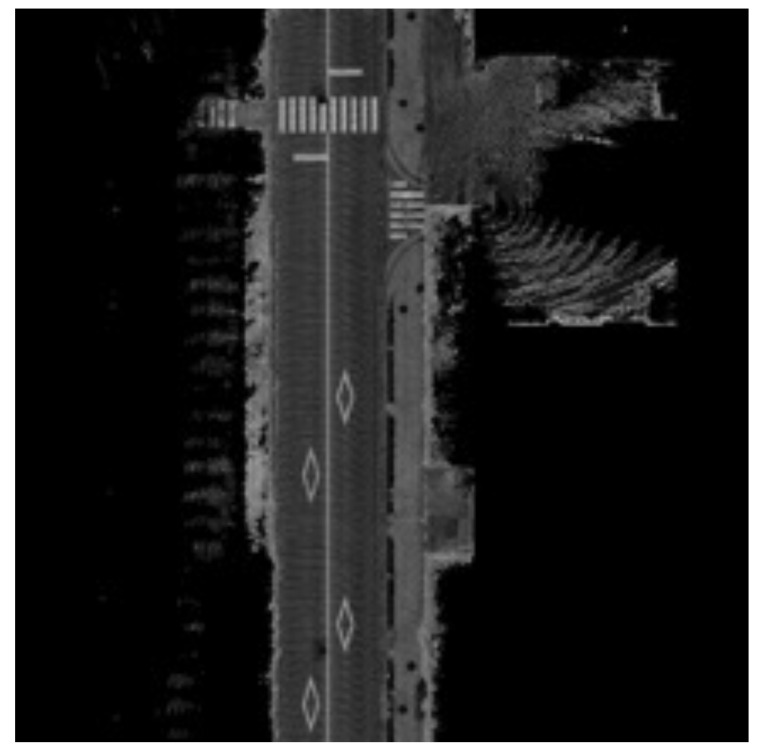
LiDAR map. The resolution of the image is 0.125 [m]. Each pixel represents an infrared reflectivity. Road surface patterns such as white lines are mapped in white because of their high reflectivity.

**Figure 7 sensors-22-03545-f007:**
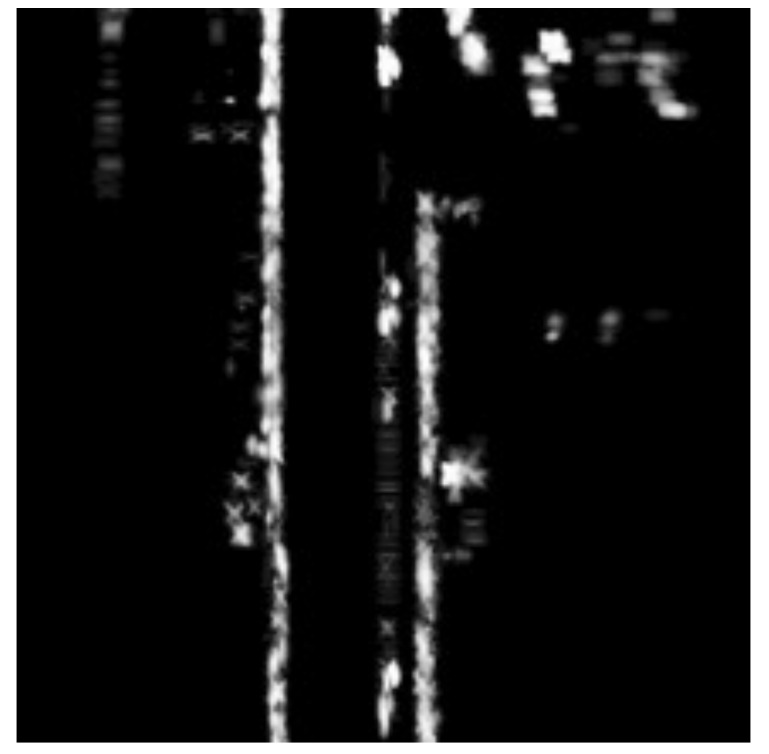
Radar map. The resolution is the same as the LiDAR map. Each pixel represents the existence probability of a stationary object at that point, with a white pixel representing a higher probability, i.e., the existence of an object.

**Figure 8 sensors-22-03545-f008:**
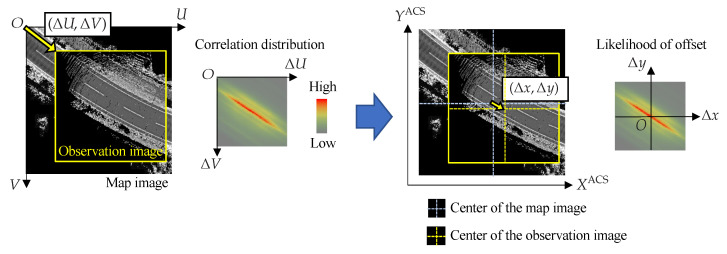
Correlation distribution and likelihood of offset.

**Figure 9 sensors-22-03545-f009:**
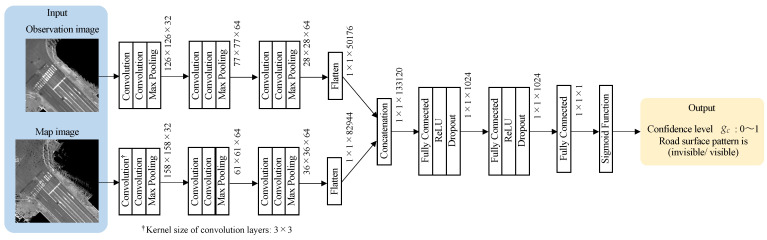
The learning structure for confidence estimation.

**Figure 10 sensors-22-03545-f010:**
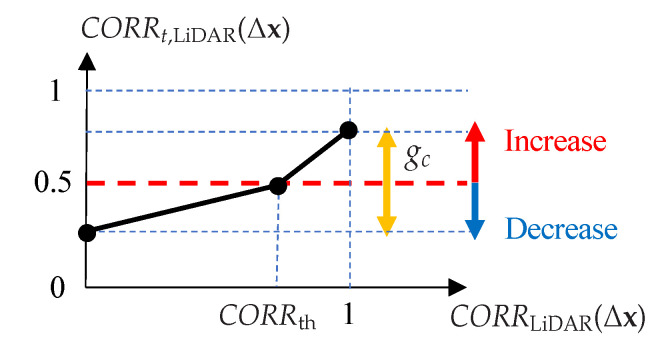
Calculating likelihood from correlation.

**Figure 11 sensors-22-03545-f011:**
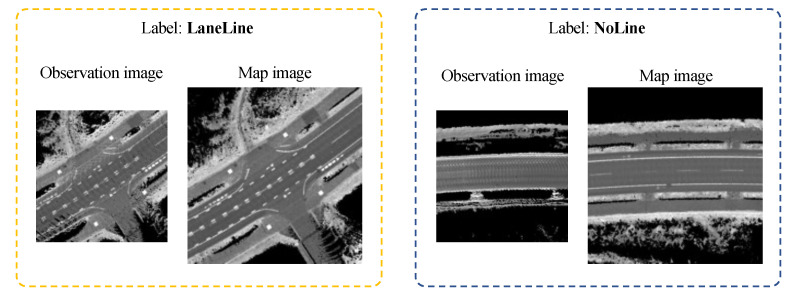
Exapmles of training data.

**Figure 12 sensors-22-03545-f012:**
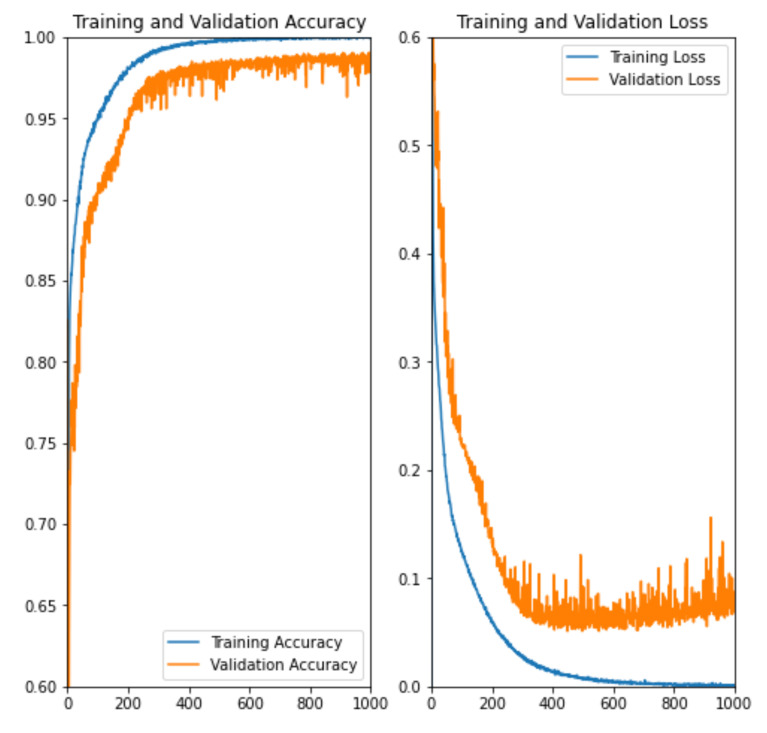
Training results. As the number of epochs increases, the accuracy and loss of both Train/Validation converge, which shows that learning has been successfully achieved.

**Figure 13 sensors-22-03545-f013:**
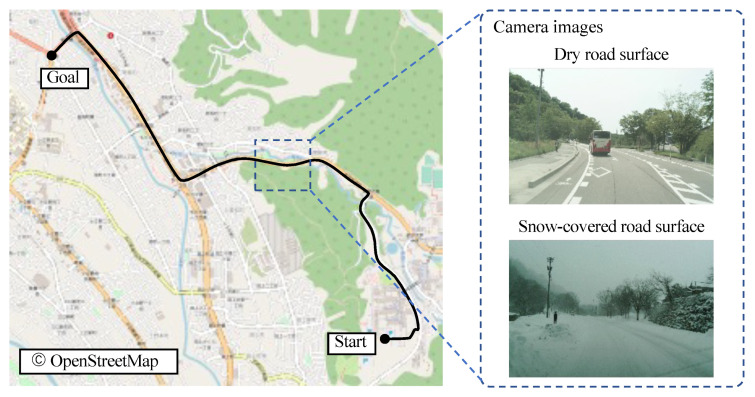
Evaluation route. The driving distance is approximately 4.6 km in one direction. Camera images are also shown as examples of actual road surface conditions under the two situations.

**Figure 14 sensors-22-03545-f014:**
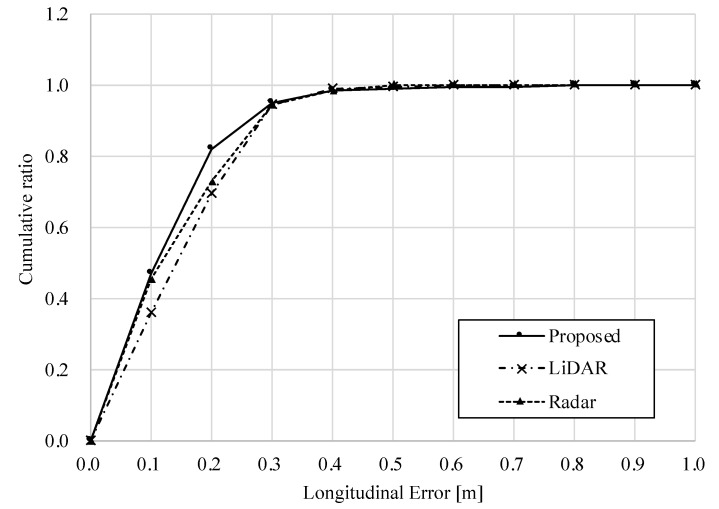
Longitudinal error in dry road surface condition.

**Figure 15 sensors-22-03545-f015:**
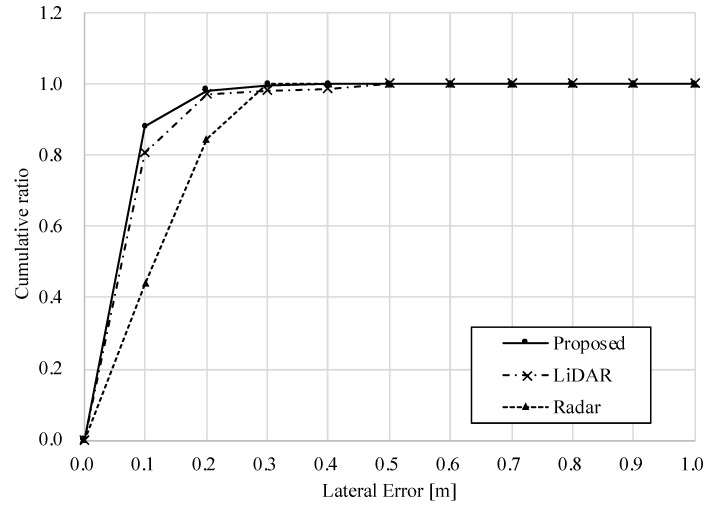
Lateral error in dry road surface condition.

**Figure 16 sensors-22-03545-f016:**
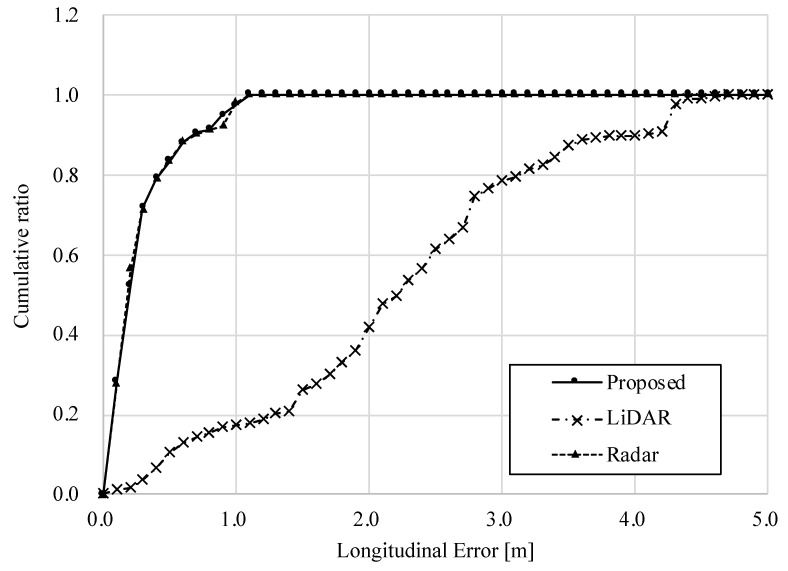
Longitudinal error in snow-covered road surface condition.

**Figure 17 sensors-22-03545-f017:**
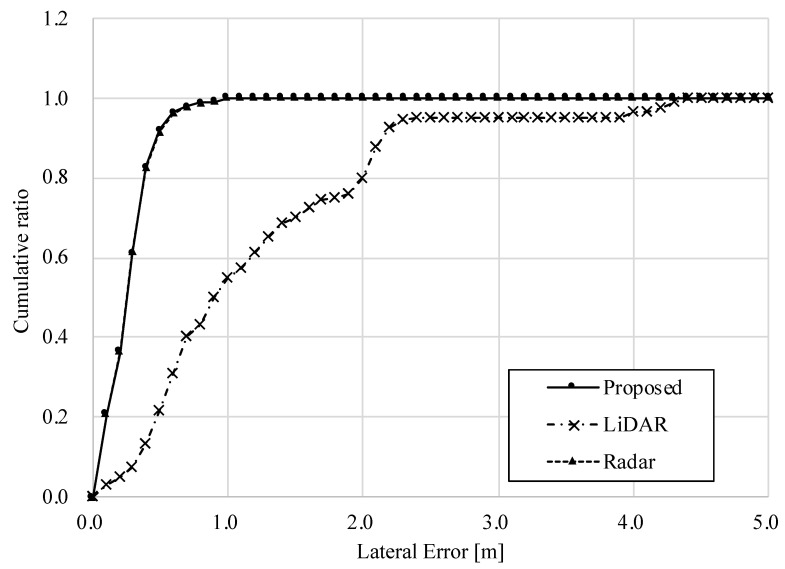
Lateral error in snow-covered road surface condition.

**Figure 18 sensors-22-03545-f018:**
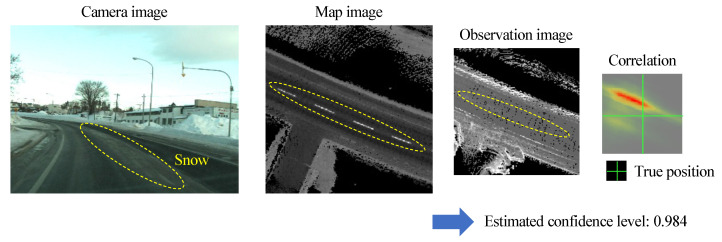
Example of incorrect confidence estimation. The area corresponding to the snow-covered region shown in the camera image is circled by a dotted line on the map and observed images. The peak of the correlation is at a different location from the true position, but a very high confidence level of 0.984 is estimated.

**Figure 19 sensors-22-03545-f019:**
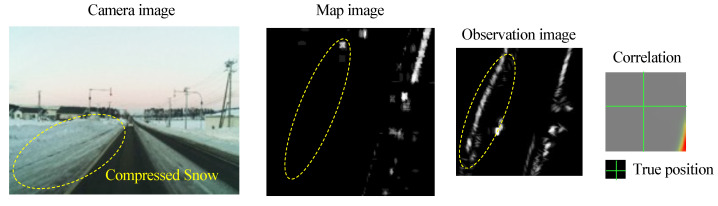
Example of incorrect matching of radar images. The area corresponding to the compressed snow shown in the camera image is circled by a dotted line in the map and observation images. It can be found that objects that do not exist in the map image appear in the observation image. Because the features between the two images are quite different, a peak of correlation appears at a location that deviated from the true position due to the matching.

**Table 1 sensors-22-03545-t001:** The number of data for training and validation.

	LaneLine	NoLine
Train	3158	5835
Validation	1392	288

**Table 2 sensors-22-03545-t002:** Loss and accuracy at the end of learning.

	Loss	Accuracy
Train	0.0013	0.9996
Validation	0.0874	0.9833

**Table 3 sensors-22-03545-t003:** Confusion matrix.

		Prediction	
		LaneLine	NoLine
Ground	LaneLine	1364	27
truth	NoLine	4	284

**Table 4 sensors-22-03545-t004:** RMS and max errors on dry road surface condition.

	Longitudinal Error [m]		Lateral Error [m]	
**Method**	**RMS**	**Max**	**RMS**	**Max**
Proposed	0.164	0.776	0.075	0.475
LiDAR	0.172	0.666	0.087	0.507
Radar	0.164	0.476	0.139	0.297

**Table 5 sensors-22-03545-t005:** RMS and max errors on snow-covered road surface condition.

	Longitudinal Error [m]		Lateral Error [m]	
**Method**	**RMS**	**Max**	**RMS**	**Max**
Proposed	0.365	1.08	0.321	0.975
LiDAR	2.38	4.75	1.54	4.36
Radar	0.360	1.05	0.318	0.971

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
