# Peer review of "LiDAR- and Radar-Based Robust Vehicle Localization with Confidence Estimation of Matching Results"

_sensors, 2022, doi:10.3390/s22093545_

Round 1

Reviewer 1 Report

You submitted an interesting research work on using the proposed confidence level to decide on whether to put more or less emphasis on LIDAR towards RADAR data, for localization in changing weather conditions. Please find below some issues and open questions:

  • [ll. 158 below] in the vehicle position in the absolute coordindate system, the y-component is also printed bold; however, it should be printed in regular font since it is not a vector
  • [ll. 158 below] the rotation matrix R is a matrix and should be typeset in bold font (typical convention)
  • [eq. (4)] descriptive subscripts such as “center” are no variables, i.e., cannot be replaced by values; hence they should be typeset using roman font and not italic
  • [ll. 159] check language/grammar of sentence
  • [ll. 174] Some reference to error determination/the error model is required here
  • [ll. 190, below] “correlation distribution” - it would make sense to already introduce the symbol used for the correlation distribution here and also use that symbol in the corresponding figures; as mentioned below, currently the symbol R is used which however has already been used for the rotation matrix
  • [ll. 207] it would be of large interest, how the architecture for the ANN has been chosen; are similar architectures used in similar/other applications?
  • [figure 9] symbol for the confidence level should already be introduced here and included in the figure as output of neural network.
  • [ll. 238 below] “The left subscript of these distributions denotes the distribution at which time …” proofread sentence, check language
  • [ll. 242 below] “This process is based on previous studies” - What does „based on previous studies“ mean? Are there any papers on that method, or is it a method which has been empirically developed internally and has not been published?
  • [symbol R_corr_lidar] Symbol R was previously used for the rotation matrix; double symbol use should be avoided.
  • [ll. 246 below] “Gamma correction is performed by cubing the correlation value as in (11)” What is the reason that radar gamma correction is different (exponent 3) to the LIDAR gamma correction (exponent 4)?

Author Response

Thank you for your comments and questions regarding the incompleteness of the manuscript.
We have corrected the formulas and grammar that you pointed out and also added explanations where they were lacking.

Reviewer 2 Report

Dear authors,

       this paper presents a visual localization system based on radar and LiDAR sensors. In particular, snowy scenes are addressed where the road surface pattern is not fully visible. Experiments have been conducted on self-collected data and the results show that the combination method can improve the localization performance, particularly in snowy scenes. The reviewer has some suggestions and comments:

  1. Please consider comparing your proposed method with some existing LiDAR-, Radar-, or sensor-fusion-based methods. This would help verify the superiority of your proposed method.
  2. The collected dataset would be a very nice contribution to the field. Would you consider making the dataset publicly available to foster future research?
  3. Since you have also gathered images with the data, would it be nice to briefly discuss some visual localization methods, especially those in Sensors.
  4. Would it be nice to add "snowy scenes" in the title? This is one of the highlights of the paper. If this is not reasonable, please ignore this suggestion, which is also perfectly fine.
  5. How about snowy semantic segmentation methods, which are also designed for autonomous driving? [*] "Improved scan matching performance in snowy environments using semantic segmentation." 2021 IEEE/SICE International Symposium on System Integration (SII). IEEE, 2021. [*] "Multi-modal sensor fusion-based semantic segmentation for snow driving scenarios." IEEE Sensors Journal 21.15 (2021): 16839-16851. [*] "Trans4Trans: Efficient transformer for transparent object and semantic scene segmentation in real-world navigation assistance." IEEE Transactions on Intelligent Transportation Systems (2022).
  6. Please consider presenting some computation complexity results. This can help verify the efficiency of the proposed method, which is important for driving applications.
  7. How is the calibration done for the sensor fusion system in your work? Please briefly describe this.
  8. The adam optimizer should be referenced when mentioning it.
  9. More descriptions on the dataset could be added. Which month did you collect the normal/snowy driving data? How were the original data processed into the current data?
  10. There are some parameter settings in your proposed method. It would be nice to conduct some parameter studies for analysis. This can help the readers better understand the work.

For these reasons, a revision is recommended.

Sincerely,

Author Response

Thank you for your helpful comments and suggestions.

We have added and corrected the points you mentioned.
As for the suggestion to release the data set, we would like to expand the data set and make it available to the public in the future.
